# Geochemical Data Mining by Integrated Multivariate Component Data Analysis: The Heilongjiang Duobaoshan Area (China) Case Study

**Zhonghai Zhao** [1,*], **Kai Qiao** [1] , **Yiwen Liu** [1], **Jun Chen** [1] **and Chenglu Li** [2]

1    College of Mining, Liaoning Technical University, Fuxin 123000, China
2    Heilongjiang Institute of Natural Resources Survey, Harbin 150036, China
*    Correspondence: zhaozhonghai@lntu.edu.cn

**Abstract:** The Heilongjiang Duobaoshan area is located at the confluence of the Great Xing'an Range and the Lesser Xing'an Range, and the area has undergone a complex magmatic and tectonic evolutionary history resulting in a complex and diverse geological background for mineralization. As a result of this geological complexity and the multi-period nature of mineralization, the geochemical data of the area are usually not satisfied with a single statistical distribution form, so traditional statistical methods cannot adequately explore and identify the distribution of deep-seated information in the geochemical data. Based on the above problems, this paper adopts a multivariate component data analysis method to process 14 mass fraction data elements, namely Ag, As, Au, Bi, Cu, Fe, Hg, Mn, Mo, Ni, Pb, Sb, W, and Zn, in the 1:50,000 soil geochemical data from the Duobaoshan area of Heilongjiang. The spatial distribution and internal structural characteristics of raw, logarithmic transformation and isometric logarithmic ratio (ILR) transformed data were compared using exploratory data analysis (EDA); robust principal component analysis (RPCA) was applied to obtain the PC1 and PC2 principal component combinations associated with mineralization, and a spectrum–area (S–A) fractal model was further used to decompose the geochemical anomalies of the PC1 and PC2 principal component combinations as composite anomalies. The results show the following: (i) The data transformed by the isometric logarithmic ratio (ILR) eliminate the influence of the original data closure effect, and the spatial scale of the data is more uniform; the data are approximately normally distributed, based on which RPCA can be applied to better explore the correlation between elements and the pattern of co-associated combinations. (ii) The S–A method was further used to decompose the composite anomalies of the PC1 and PC2 principal component combination in the study area. The anomalous and background fields of the screened-out PC1 and PC2 principal component combinations reflect anomalous information on mineralization dominated by Au mineralization. Moreover, the anomaly and background information after extraction were in good agreement with the known Au deposits (points), and many geochemical anomalies with prospecting potential were obtained in the periphery, providing a theoretical basis and exploration focus for the next step in the searching and exploring of the study area.

**Keywords:** component data; isometric logarithmic ratio transformation; robust principal component analysis; spectrum–area fractal model; Duobaoshan region

## 1. Introduction

Geochemical exploration methods have been dominant in mineral exploration and the quantitative prediction of mineral resources. Since the 1970s, geologists in various countries have accumulated a large amount of multi-scale and multi-element geochemical data in the process of mineral exploration [1–7], of which geochemical data processing is particularly important, indispensable and decisive in reconnaissance geochemistry [8–11]. Zuo et al. [12] pointed out how to efficiently process geochemical survey data and emphasized that mining and identifying deep information in the past, present, and future

has been a hot and cutting-edge area of research in survey geochemistry. In the current research, several geochemical data processing methods were proposed for spatial pattern recognition and anomaly extraction in the geochemical data models of surveys, such as conventional statistical analysis [13], local Gap statistical methods [14], multivariate statistical analysis [15], data exploration analysis [16], geo-statistics [17], and fractal as well as multifractal methods [18], among which the fractal and multifractal theory of anomaly identification and extraction is a processing method that has been proposed and developed rapidly as well as effectively in recent years.

Mandelbrot [19] created the concept of "fractal geometry" in 1983. This concept has been applied to the analysis of complex phenomena [20,21], followed by several related studies suggesting that the spatial distribution and frequencies of geochemical elements may obey self-similarity in fractal models [22–24]. A series of fractal models has been proposed for the extraction of geochemical anomalies; common fractal and multifractal methods include the following: local singularity methods [25], the concentration–area (C–A) fractal model [26], the concentration–volume (C–V) fractal model [27], the spectrum–area (S–A) fractal model [28], the number of feature spaces–eigenvalues (N–λ) fractal model [29], and the Walsh space counterpart fractal model [30]. These methods not only consider the distribution of the deep-level information of geochemical fields, but also consider the spatial correlation, geometric patterns, and scale invariance, thus enabling the effective decomposition of complex in addition to deep background and superimposed anomalies in composite geochemistry. Cheng [25] pointed out that, by studying and quantitatively analyzing geochemical data with fractal and multifractal methods, weakly retarded geochemical anomalies that are difficult to identify can be extracted from complex geological conditions. The study and quantitative analysis of fractal and multifractal geochemical data can extract information on weakly slowed geochemical anomalies that are difficult to identify from complex geological conditions, thus enabling the understanding of the geochemical element distribution patterns. Currently, fractal and multifractal methods are widely used in exploration and geochemical data processing [15,18,31–33].

With the development of digital earth science, the requirements for the ability to identify geochemical anomalies are gradually increasing. Aitchison [34] proposed the method of log–ratio transformation to improve geochemical anomaly recognition capability, as geochemical data are typically compositional data and the geometric space of compositional data is Aitchison space [35]. In the geochemical data processing methods of surveys, as mentioned above, their geometric space is based on Euclidean space, so the use of log-ratio transformations to transform the component data into the corresponding space can assist in more accurately identifying and decomposing geochemical anomalies [28]. Many studies have shown [32,36,37] that the log-ratio transformation method can reveal the true spatial distribution patterns of elements more effectively, and the elimination of the closure effect of survey geochemical data by log-ratio transformation has gradually become an important step in survey geochemical data processing. The common log-ratio transformation methods include additive log-ratio (ALR) transformation, centered log-ratio (CLR) transformation and isometric log-ratio (ILR) transformation.

The Duobaoshan area in Heilongjiang is located at the northern end of the East Ujimqin Banner–Nenjiang polymetallic metallogenic belt, a shallow cut zone in the low to medium mountains. After recent decades of searching for minerals, several deposits (points) with a high mineralization potential have been identified in and around the area (Figure 1). Due to the geological complexity of the area and the multi-phase nature of mineralization, geochemical data are usually not satisfied with a single statistical distribution form, so traditional statistical methods are not well suited to uncover and identify the distribution of deeper information in geochemical data. Therefore, in this study, 14 elements, Ag, As, Au, Bi, Cu, Fe, Hg, Mn, Mo, Ni, Pb, Sb, W, and Zn in the 1:50,000 soil geochemical data from the Duobaoshan area of Heilongjiang were processed based on multivariate component data analysis, and the spatial distribution as well as the internal structural characteristics of the original data, logarithmic data, and ILR-transformed data were compared using exploratory

data analysis (EDA). Robust principal component analysis (RPCA) was applied to obtain the PC1 and PC2 principal component combinations associated with the mineralization of the study area. The spectrum–area(S–A) fractal model was further used to decompose the composite anomalies of the combined PC1 and PC2 principal component geochemical anomalies, thus revealing the true spatial distribution pattern of the geochemical elements in the study area more effectively as well as providing ideas and directions for the further search for minerals in the area.

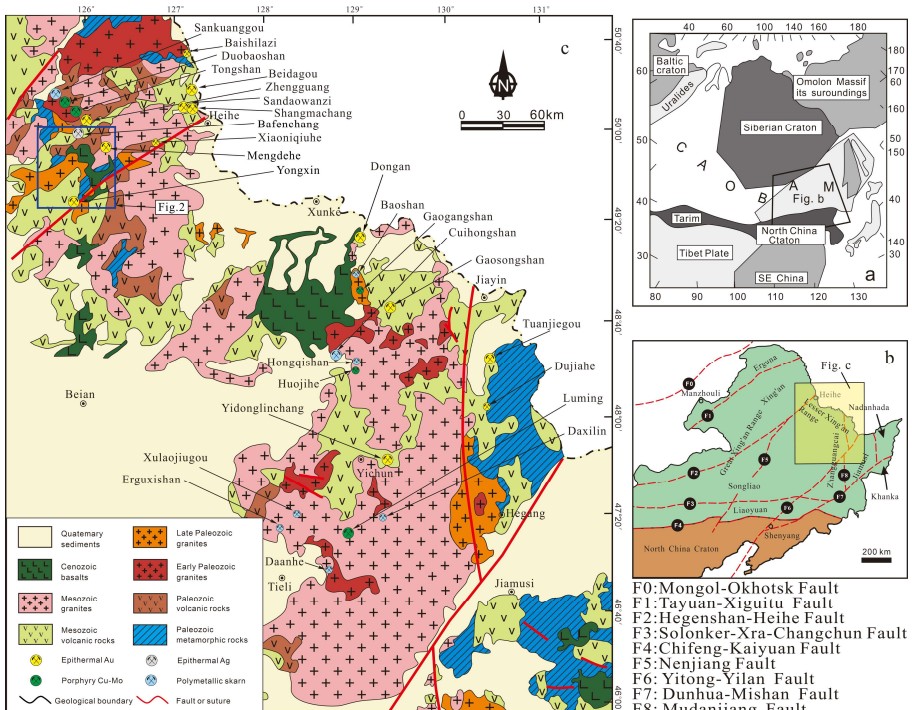

**Figure 1.** (**a**,**b**) Tectonic divisions of Northeast China and (**c**) a regional geological map of the Lesser Xing'an Range.

## 2. Geological Profile

### 2.1. Regional Geological Background

The study area is located at the eastern end of the Central Asian Orogenic Belt (Figure 1a), at the confluence of the Great Xing'an Range and the Lesser Xing'an Range in Heilongjiang (Figure 1b). The northeastern Great Xing'an Range is a superimposed complex tectonic zone that has undergone a long and complex magmatic and tectonic evolution, with a complex series of microplate fits between the Siberian and North China plates in addition to the tectonic evolution and eventual closure of the Paleo-Asian Ocean since the Palaeozoic [38–40]. The Lesser Xing'an Range area is characterized by the development of volcanic, metamorphic, and granitic rocks of different ages, and is the most complex and intense area of tectonic–magmatic evolution in the northeast [41–43]. In recent years, with the continuous exploration of minerals in the region, various large- and medium-sized deposits have been discovered, such as porphyry copper–molybdenum deposits in Tongshan [44] and DuobaoShan [45]; silica-type iron–copper (molybdenum) deposits, such as in Cuihongshan [46] and Xulaojiugou [47]; and shallow-forming low-temperature hydrothermal Au deposits, such as in Zhengguang [48], Sandaowanzi [49], and Tuanjiegou [50] (Figure 1c). This shows that this area has excellent metallogenic potential and a good metallogenic geological background. Up to the present, a Yongxin Au deposit, a Mengdehe Au deposit, a Bafenchang Ag-Au deposit, and many other mineralization points have been found in the study area (Figures 1 and 2).

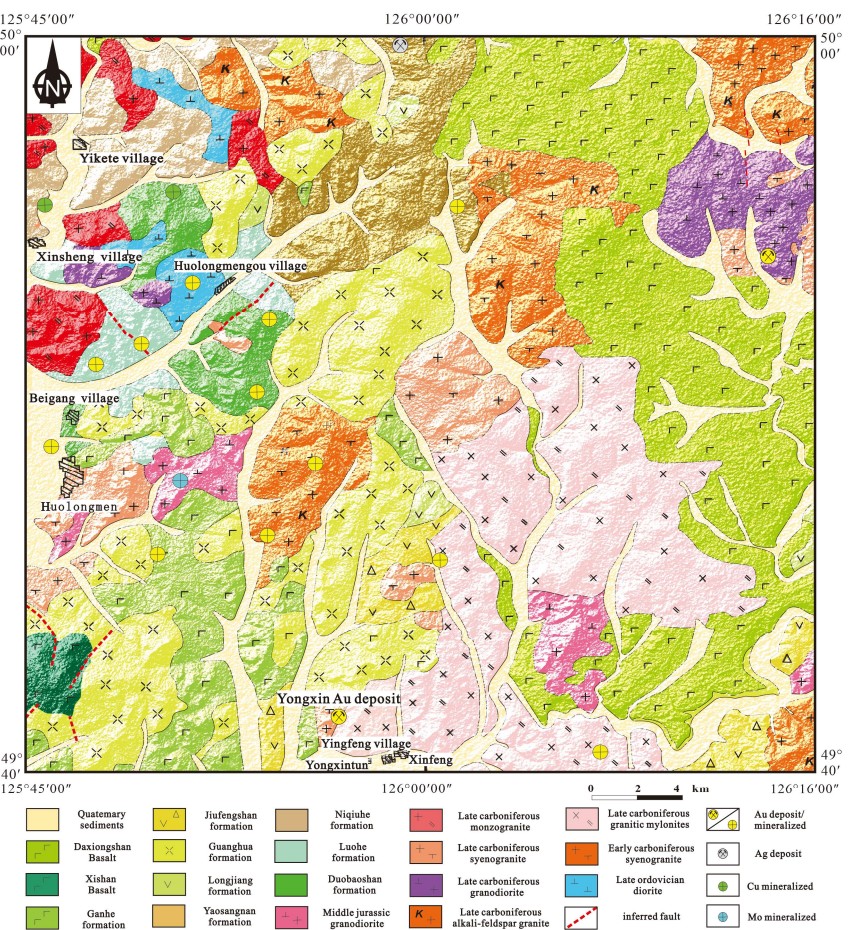

**Figure 2.** Geological sketch of the study area.

## 2.2. Geological Background of the Study Area

The stratigraphic units exposed in the study area are numerous and widely distributed, mainly including Palaeozoic, Mesozoic, and Cenozoic strata. The Palaeozoic strata are widely distributed in the north-western part of the area, the Mesozoic strata are concentrated in the south-western part of the area, and the Cenozoic strata are mainly distributed in the eastern part of the area. The Palaeozoic strata in the region include the Duobaoshan Formation ($O_{1-2}d$), the Luohe Formation ($O_3l$), the Niqiuhe Formation ($S_3D_2n$), and the Yaosangnan Formation ($D_2y$); the Mesozoic strata include the Longjiang Formation ($K_1l$), the Guanghua Formation ($K_1gn$), the Jiufengshan Formation ($K_1j$), and the Ganhe Formation ($K_1g$); and the Cenozoic strata in the region are mainly Quaternary high and low river floodplain deposits ($Qh$) (Figure 2). The intrusive rocks are widely distributed, and the rock types are complex, ranging from neutral to acidic rocks, with granites of medium-to-deep formations predominating. The formation age is in the following order: Middle Ordovician, Carboniferous, Middle Jurassic, and Early Cretaceous. The Palaeozoic Duobaoshan Formation, Mesozoic Guanghua Formation, Longjiang Formation, and Early-to-Late Carboniferous granites are the host rocks for several significant Au and polymetallic ore deposits, including the Zhengguang Au deposit, Sandaowanzi Au deposit, and Yongxin Au deposit [45,51,52]. The alteration of the surrounding rocks of the deposit mainly includes actinolitization, chloritization, epidotization, propylitization, silicification, sericitization, carbonatization, pyritization, etc. Most of the deposits (sites) are located at the intersection of NE-oriented and secondary NW-oriented fracture structures, and hydrothermal alterations such as silicification, pyritization, chloritization, and carbonatization are commonly developed.

## 3. Methods

### 3.1. Data Collection and Analysis

The geochemical data are from the 1:50,000 soil geochemical survey in the "Heilongjiang Duobaoshan Area Mineral Vision Survey Project", covering an area of 1334 km$^2$, with 10,314 soil samples collected. Sample collection strictly adhered to the requirements of the Geochemical Census Specification (1:50,000) (DZ/T0011-2015). The soil samples for this soil survey were collected at 333 × 333 m designed sampling points with a sampling point error of less than 15 m, and sampling was taken point-by-point by line. The sample sampling density was 9 points/km$^2$ of grid sampling. The sampling layer is the deposition layer (B layer), the sampling depth was 30–80 cm, mostly 40–50 cm, and the sampling media were sand, clay, and subclay. Each sample was delivered at a weight of 150 g and sieved with a −10 to +60 mesh. Samples were processed to 200 mesh by fine-grinding the collected samples according to the requirements specified by the laboratory. To prevent sample contamination, the preparation of samples using a nonpolluting sample grinder and partially by hand-grinding in an agate emulsion bowl, and the processing of the chemical probe samples, needed to be completely separate from the processing of the ore in the laboratory.

This sample was analyzed and tested by the Testing Centre of the Heilongjiang Geological Survey Research Institute and the Testing Centre of the Qiqihaer General Institute of Mineral Exploration and Development. Fourteen elements, Ag, As, Au, Bi, Cu, Fe, Hg, Mn, Mo, Ni, Pb, Sb, W, and Zn, were quantitatively analyzed and tested, and the analytical methods [53] and detection limits of the elements are shown in Table 1.

**Table 1.** Analysis methods and parameters.

| Element | Analysis Method | Detection Limit | Precision (RSD%) |
|---|---|---|---|
| Ag | ES | 0.02 mg/kg | 5.32 |
| As | AFS | 0.20 mg/kg | 4.98 |
| Au | GAAS | 0.30 mg/kg | 4.39 |
| Bi | AFS | 0.03 mg/kg | 4.71 |
| Cu | XRF | 1.00 mg/kg | 4.54 |
| Hg | AFS | 0.01 mg/kg | 6.72 |
| Mn | XRF | 5.60 mg/kg | 2.49 |
| Mo | ES | 0.24 mg/kg | 6.99 |
| Ni | XRF | 2.80 mg/kg | 1.31 |
| Pb | XRF | 1.50 mg/kg | 3.67 |
| Sb | AFS | 0.05 mg/kg | 5.37 |
| W | POL | 0.31 mg/kg | 4.92 |
| Zn | XRF | 3.00 mg/kg | 2.59 |
| Fe | XRF | 0.05 mg/kg | 2.22 |

Note: AFS—atomic fluorescence spectrometry; ES—emission spectrography; GAAS—gallium arsenide; POL—polarography; and XRF—X-ray fluorescence.

### 3.2. Data Processing

3.2.1. Log-Ratio Transformation and Robust Principal Component Analysis

Since the 1980s, a large number of mathematical geologists have begun to establish methods and theories for compositional data analysis [35,54]. Since geochemical data are distributed in a limited area and obey units as well as constraints, they are typical compositional data. The sum of all elemental contents in compositional data is a constant value known as the "closure effect". The closure effect can lead to pseudo-correlations between geochemical variables, making the results of data processing methods based on correlations between elements uncertain. Most compositional data do not follow the characteristics of a normal distribution; in traditional geochemical studies, log-transformation is often chosen to make them follow the characteristics of a normal distribution. However, the closure effect affects the log-transformed data and cannot obtain mineralogical element combinations with a clear indication of mineralization. Aitchison [35] proposed the ALR and CLR trans-

formations to overcome the effects of closure in compositional data. Egozcues et al. [55] proposed the ILR transformation, which is based on the assumption that the sample space is given Euclidean geometry. The log-ratio transformation addresses the effects of closure in compositional data by transforming the original data (e.g., geochemical data) from the geometry of the compositional data into Euclidean space [7].

Compared with the non-isometric transformations, ALR and CLR, the variables in the Euclidean space will change after the transformation; the ILR transformation can ensure that the relative distance between the variables before and after the transformation of the component data remains unchanged [13], and is more suitable for the processing of component data. However, this method is asymmetric, and the correspondence between the variables will be disrupted after the ILR transformation, so the transformed variables cannot be directly interpreted. Filzmoser proposed a combination of the ILR transformation and robust principal component analysis (RPCA) [56–58], in which the ILR transforms the original data which are then analyzed using the RPCA method to obtain the principal component scores as well as loadings and then converted them into a CLR coordinate system by using a standard orthogonal basis inverse. The data are then transformed into the CLR coordinate system to establish the relationship with the original variables. The problem of mismatched variables after the ILR transformation is thus solved. Moreover, compared with the traditional principal component analysis, RPCA is based on robust statistics, which can suppress the influence of outliers in geochemical data on the results of the principal component analysis by constructing a robust covariance matrix or correlation matrix. The present study involves the ILR transformation formula as shown below:

$$CLR(x) = ln\frac{x_i}{\left\{\prod_{j=1}^{D} x_j\right\}^{\frac{1}{D}}}, i = 1, 2, \cdots, D$$

$$ILR(x) = \sqrt{\frac{i}{i+1}}ln\frac{\sqrt[i]{\prod_{j=1}^{i} x_j}}{x_{i+1}}, i = 1, 2, \cdots, D-1$$

### 3.2.2. Spectrum–Area Fractal Model

Geochemical data are generally influenced by elements such as mineralization and regional geology, giving them the advantage of complexity and diversity. Cheng et al. [59] proposed a fractal method for decomposing composite and superimposed anomalies developed based on the principle of generalized self-similarity, which is also known as the spectrum–area (S–A) fractal model or fractal filtering method. The geochemical field obeys self-similarity between indices and scales, and specific geological processes or phenomena of spatial relevance usually respond to the fraction with self-similarity. In frequency domain space, the S–A method is based on this self-similarity to construct a fractal filter and invert the fractal-filtered information back into the spatial domain using a Fourier transform transformation to obtain the decomposed background and anomaly maps. The S–A expression is shown below [28]:

$$A(\geq S) \propto S^{-\beta}$$

where S reflects the spectrum density; A(≥S) reflects the area in the spatial region where the energy spectrum density is greater than S; and β is the exponential factor. When the energy spectral density (S) is larger than the spectral density (A(≥S)) in the spatial region obeying the exponential relationship, then they are simultaneously taken logarithmically and plotted on a logarithmic graph. On the plotted lnS-lnA(≥S), different linear and fractal relationships can be reflected according to the differences between the intervals where S is taken and the differences between the straight-line segments while using the intervals. The distribution of the demarcation points is used to determine the threshold of the fractal filter [60,61].

## 4. Results and Discussion

### 4.1. Multivariate Component Data Analysis

The elemental analysis results of the 10,314 soil samples collected within the study area were analyzed using SPSS and R software. The geochemical characteristics of the 14 elements were discussed according to their means and standard deviations (Table 2), and the raw data, log-transformed data, and ILR-transformed data were statistically analyzed to identify statistical patterns between the elements. An exploratory data analysis (EDA) method was also used to visualize the data using box plots, density histograms, and double-labeled plots so that the internal structure and dispersion characteristics of the data could be obtained quickly and accurately.

**Table 2.** Statistics of the raw data, logarithmically transformed data, and isometric log-ratio transformation data of samples from the study area.

| Element | | Ag | As | Au | Bi | Cu | Hg | Mn | Mo | Ni | Pb | Sb | W | Zn | Fe |
|---|---|---|---|---|---|---|---|---|---|---|---|---|---|---|---|
| Minimum | | 0.04 | 1.00 | 0.10 | 0.05 | 3.20 | 0.01 | 85.00 | 0.25 | 3.10 | 9.00 | 0.04 | 0.27 | 15.00 | 0.59 |
| percentiles | 25% | 0.07 | 8.30 | 0.70 | 0.30 | 18.90 | 0.03 | 658.00 | 0.80 | 22.40 | 23.20 | 0.49 | 1.76 | 62.70 | 3.57 |
| | 50% | 0.08 | 9.80 | 1.00 | 0.34 | 22.50 | 0.03 | 972.00 | 0.96 | 25.70 | 25.60 | 0.57 | 1.96 | 73.50 | 3.96 |
| | 75% | 0.10 | 12.10 | 1.50 | 0.38 | 25.90 | 0.04 | 1228.00 | 1.18 | 29.60 | 28.40 | 0.69 | 2.17 | 84.60 | 4.34 |
| Maximum | | 3.58 | 151.20 | 1309.50 | 23.04 | 193.50 | 1.22 | 6865.00 | 108.90 | 194.20 | 228.60 | 12.89 | 50.86 | 347.00 | 8.67 |
| Std | | 0.08 | 5.35 | 17.60 | 0.36 | 8.57 | 0.02 | 455.00 | 1.50 | 8.85 | 5.94 | 0.30 | 1.11 | 18.80 | 0.63 |
| Mean | | 0.09 | 10.70 | 1.74 | 0.37 | 23.40 | 0.03 | 977.00 | 1.11 | 26.70 | 26.20 | 0.62 | 2.06 | 75.30 | 3.93 |
| Raw | Skew | 17.77 | 9.22 | 66.50 | 37.28 | 4.82 | 22.19 | 1.42 | 42.38 | 5.20 | 8.55 | 13.11 | 22.75 | 2.32 | −0.27 |
| | Kurt | 547.06 | 169.66 | 4655.68 | 1966.62 | 55.15 | 930.34 | 9.08 | 2682.53 | 68.48 | 195.55 | 379.33 | 765.61 | 17.65 | 2.28 |
| | MAD | 0.02 | 2.67 | 0.59 | 0.06 | 5.19 | 0.01 | 418.09 | 0.28 | 5.34 | 3.85 | 0.13 | 0.31 | 16.16 | 0.58 |
| log10 | Skew | 2.27 | 0.44 | 1.41 | 2.13 | 0.16 | 0.64 | −0.76 | 2.04 | −0.17 | 1.17 | 0.96 | 1.68 | 0.16 | −1.58 |
| | Kurt | 9.72 | 6.05 | 7.40 | 19.70 | 4.03 | 5.89 | 0.60 | 13.43 | 5.32 | 9.64 | 7.87 | 19.51 | 2.63 | 7.41 |
| | MAD | 0.10 | 0.11 | 0.23 | 0.07 | 0.10 | 0.13 | 0.18 | 0.13 | 0.09 | 0.06 | 0.11 | 0.07 | 0.10 | 0.06 |
| ILR | Skew | 1.78 | 0.50 | 1.45 | 2.43 | 0.66 | 0.77 | −0.73 | 1.86 | 0.67 | 0.12 | 0.94 | 1.87 | 0.41 | −0.66 |
| | Kurts | 7.32 | 4.38 | 8.30 | 20.50 | 4.11 | 5.52 | 0.42 | 12.05 | 6.63 | 2.30 | 8.69 | 15.45 | 0.85 | 4.22 |
| | MAD | 0.24 | 0.26 | 0.49 | 0.15 | 0.19 | 0.28 | 0.44 | 0.28 | 0.17 | 0.18 | 0.22 | 0.16 | 0.23 | 0.15 |

Note: Std—standard deviation. Skew—skewness. Kurt—kurtosis. MAD—median absolute deviation. All of the element content values are expressed in an exponential form but with the exponential part ($10^{-9}$ for Au and $10^{-6}$ for all the other elements) omitted from the Table for convenience.

The median absolute deviation (MAD) value reflects the median of the absolute value of the new data obtained by subtracting the median from the original data [62]. The mean and standard deviation of the data is susceptible to outliers and is less stable, whereas the MAD based on robust statistics is less affected by outliers, more stable, and more accurate in presenting the data center and the degree of dispersion of the data. In Table 2, it can be seen that the MAD values of the raw data are highly variable in the mean and standard deviation of the elemental content. This result also reflects the difference in the spatial distribution of various elements due to the influence of various geological elements in the study area.

From Table 2 and the box plot (Figure 3a), it can be seen that the skewness and kurtosis of the raw dataset are too high to meet the requirements of a normal distribution; the spatial scale of the raw dataset is large, the data are scattered, and some elements have a large number of high-value discrete points. Compared to the raw dataset, the log-transformed and ILR-transformed data (Table 2 and Figure 3b,c) show substantially less variability in the scale of the distribution of the elements, and the data for each element lie essentially at the same order of magnitude; the elements' skewness and peak state were significantly improved, with the bias and kurtosis of the elements closer to the normal distribution. The density curve (Figure 4) also shows a single-peaked distribution. In contrast, the original data do not show a corresponding density curve due to the large differences in scale; the transformed elemental data are homogeneous, and the data tend to be centered, which is more in line with the requirements of multivariate statistical analysis.

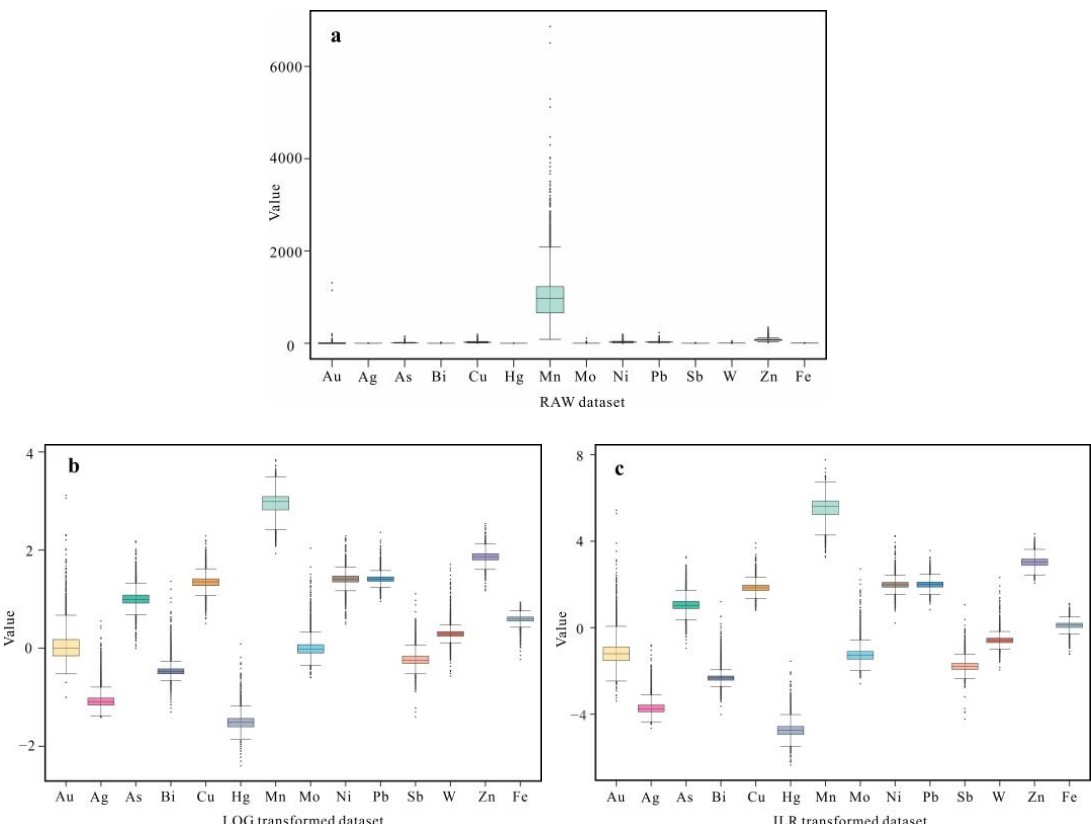

**Figure 3.** Box plots of (**a**) raw, (**b**) log-transformed, and (**c**) ILR-transformed datasets of metallogenetic elements.

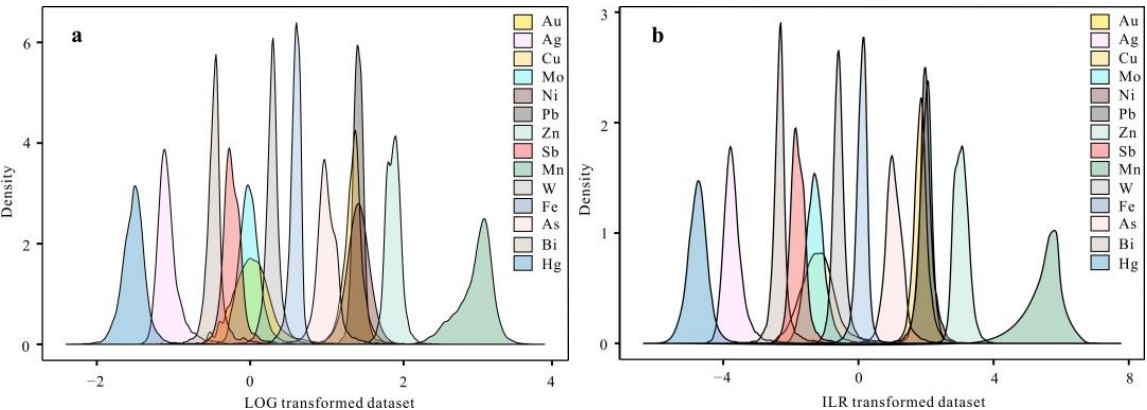

**Figure 4.** Density curves of (**a**) log-transformed and (**b**) ILR-transformed datasets of metallogenetic elements.

In order to better explore the correlation and co-association patterns among the elements in the study area, the principal component analysis (PCA) results are visualized in this study by combining the label plots of the EDA method. From Figure 5a,b, it can be seen that the original data and the log-transformed data have positive loadings on all of the elements in the PC1 principal component, and no information on the correlation between the elements can be derived. In contrast in the PC2 principal component, the two only show minor differences, and the results of the principal component analysis are bounded by the data closure effect. For the data based on the ILR transformation (Figure 5c,d), after PCA and RPCA, the variables were radioactive, the closure effect was obviously eliminated, and the relationship between the transformed data was much clearer. They also indicate that the principal components obtained by this method are more representative, with Au,

As, Cu, Fe, Hg, Pb, and Sb showing positive loadings in the PC1 principal component and Au, Ag, Mn, and Zn showing positive loadings in the PC2 principal component, with Au being the most discrete element as well as having high loadings in both the PC1 and PC2 principal components, indicating that Au is the main-ore forming metal, and that the PC1 and PC2 principal components may reflect a combination of elements associated with Au mineralization.

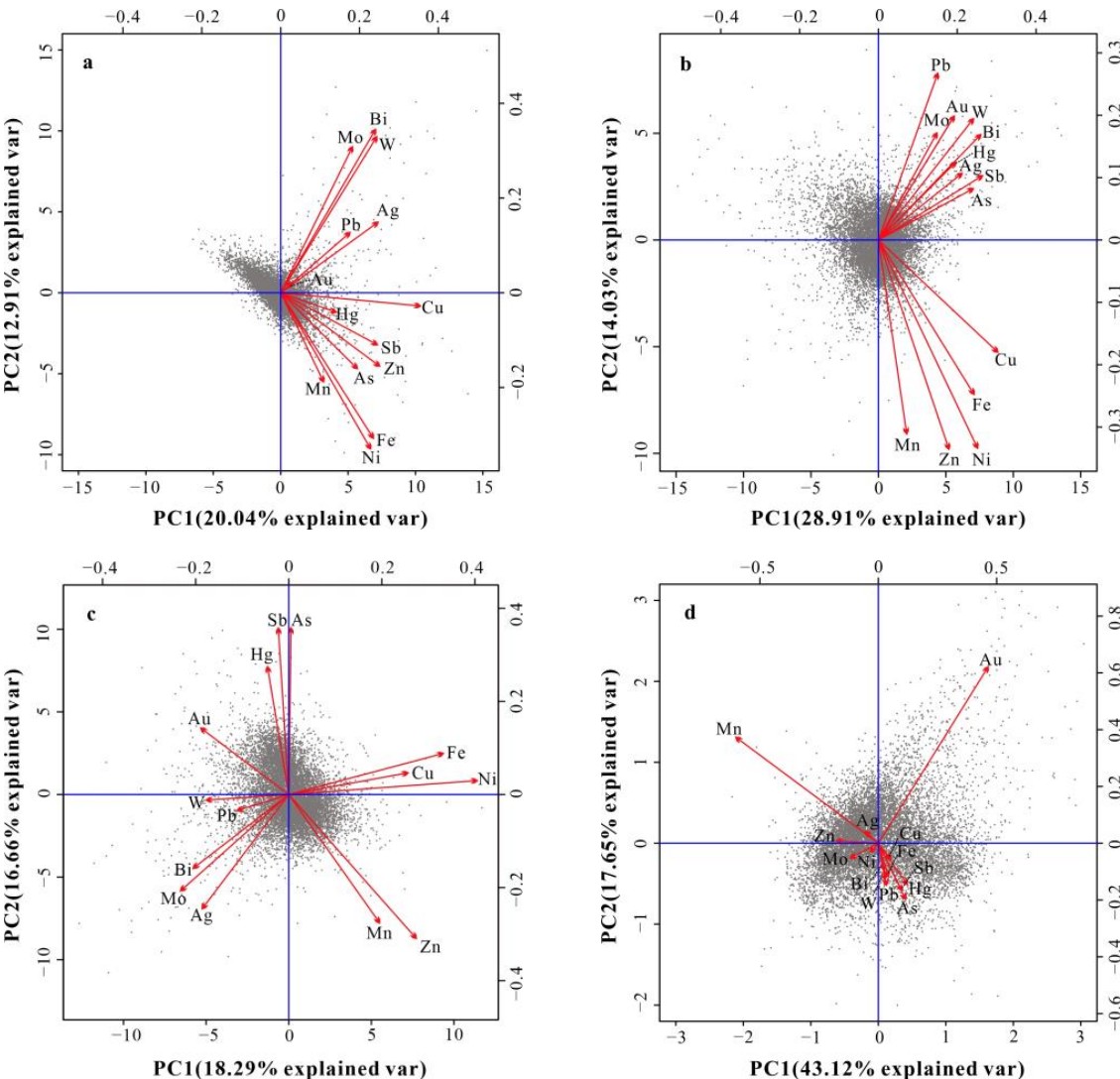

**Figure 5.** Biplots of the PC1 and PC2 were obtained by the raw dataset with PCA (**a**) and log-transformed dataset with PCA (**b**) and ILR-transformed dataset with PCA (**c**) and ILR-transformed dataset with RPCA (**d**).

Based on the biplot analysis, this study will plot the principal component score point plots for the PC1 and PC2 principal component s' score data obtained from the RPCA using ILR transformed data. According to the principal component score point map (Figure 6), it can be seen that the PC1 principal component scores in the western part of the study area are lower compared to the eastern part. In contrast, the higher scoring areas in the east are concentrated over the Daxiongshan Basalt, indicating that there may be a higher background of influence by the Daxiongshan Basalt in this area. At the same time, this high background generated by the eastern Daxiongshan Basalt may inhibit the identification of weak anomalies in the western part of the study area. In the PC2 principal component score, the high-scoring zones are relatively discrete, mostly overlying Early Cretaceous volcanic–subvolcanic rocks and Carboniferous granites with good coincidence

and spatial correlation with known deposits (points), possibly as a result of magmatic–hydrothermal mineralization.

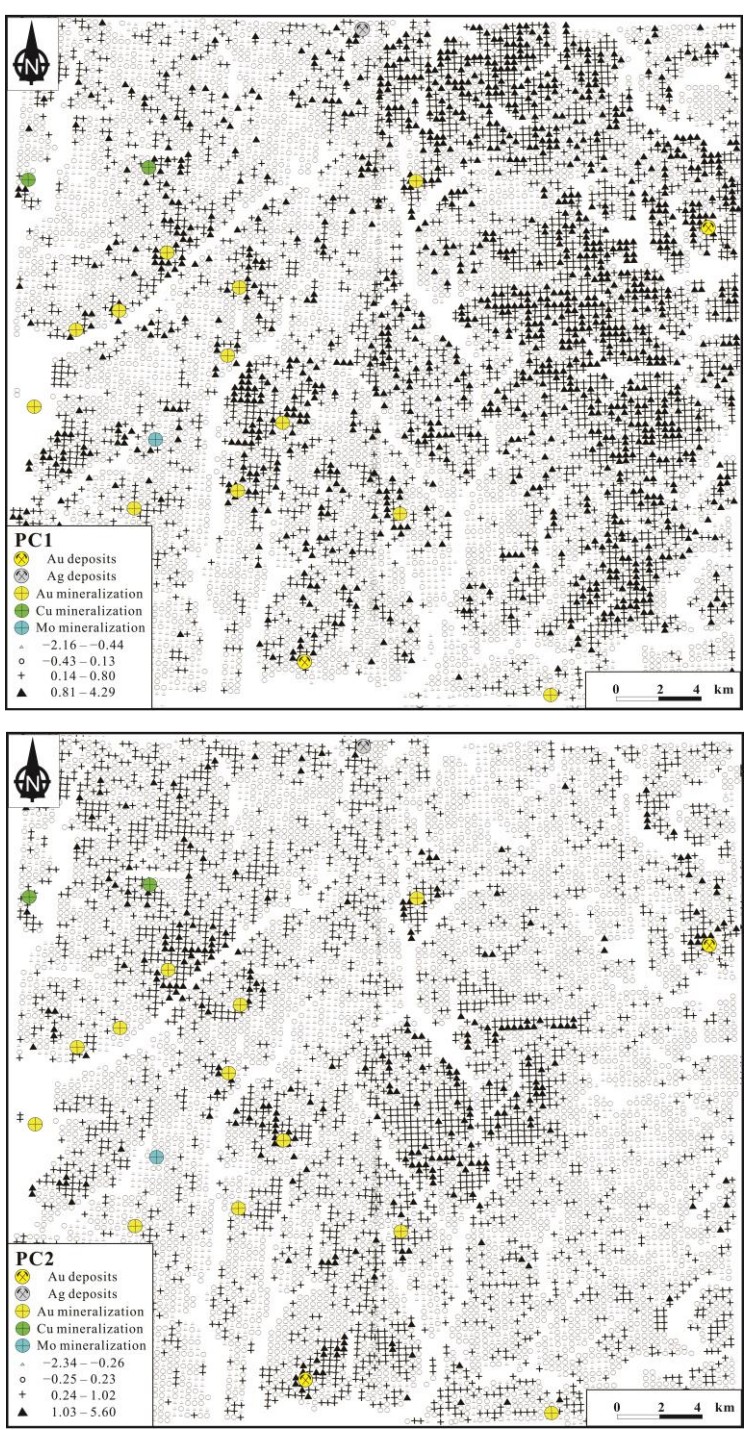

**Figure 6.** Maps showing the scores of the first component and second component of RPCA.

### 4.2. Spectrum–Area Fractal Model Analysis

In order to eliminate the influence of factors such as mineralization and regional geology and thus more accurately decode the geochemical anomaly information, the PC1 and PC2 principal components' data from the ILR transformation–based RPCA score will be further selected for kriging interpolation processing and S–A decomposition. In this study, the S–A model is implemented with the Geodas quantitative mineral resource prediction system developed by the China University of Geosciences. The data of the PC1

and PC2 principal components' scores were transformed into the frequency domain by Fourier transform, and the relationship between the energy spectral density (S) and the cumulative area (A) was obtained on the plotted lnS-lnA($\geq$S) curve. According to the variation pattern of both, a line fit based on fewest squares was performed to divide the energy spectral density values of the data into different value intervals according to the slope of the fit (number of sub-dimensions). In the lnS-lnA($\geq$S) plot of the PC1 principal component (Figure 7a), the line y = $-2.09$x + 17.10 represents the noise field, the line y = $-1.69$x + 14.98 represents the anomaly field, and the line y = $-1.55$x + 14.06 represents the background field. In the lnS-lnA($\geq$S) plot of the PC2 principal component (Figure 7b), the line y = $-2.21$x + 17.05 represents the noise field, the line y = $-1.76$x + 14.86 represents the anomalous field, and the line y = $-2.10$x + 17.05 represents the background field.

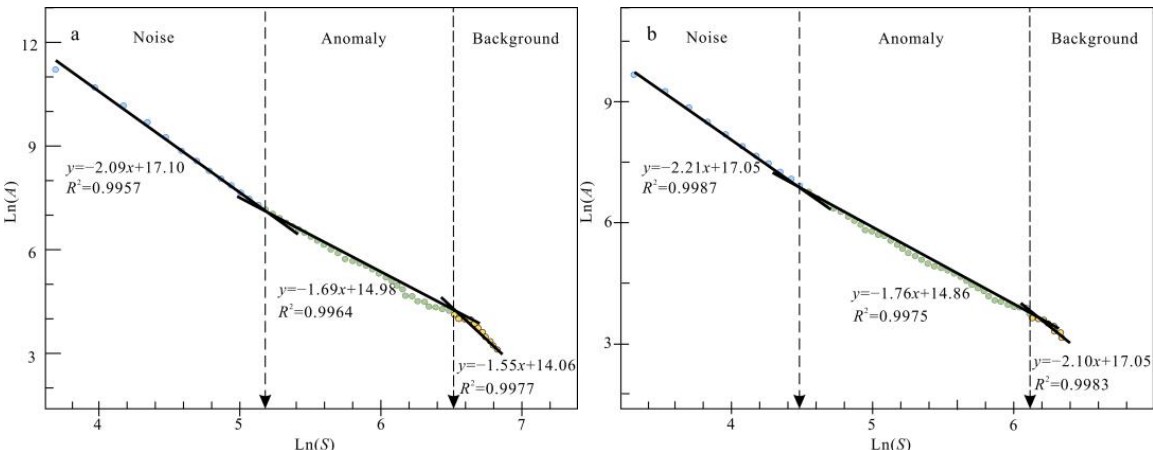

**Figure 7.** Log-log plots of power spectrum value S versus the area with power spectrum greater than S for principal component of log=ratio dataset of PC1 (**a**) and PC2 (**b**).

The background field obtained by S–A decomposition mainly reflects the background composition of elemental mass fractions; high-background areas may be favorable for polymetallic mineral exploration; variations in the background strength reflect the presence of elements in a favorable geological context for mineralization, and the anomaly field mainly reflects local anomalous mass molecules of elements and noise generated during data processing. Accordingly, the background and anomaly maps corresponding to the PC1 and PC2 principal components of the RPCA score were drawn based on the anomalies and background fields defined above. After decomposition, the background map of the PC1 principal component (Figure 8b) reflects the differences between the east and west of the Duobaoshan area, and, combined with the geological conditions of the study area (Figure 2), it can be seen that the high-background area is located above the Daxiongshan Basalt. In contrast, the known deposits (points) are located above the low-background area. The remaining anomalies of the PC1 principal component (Figure 8a) were obtained after the background anomalies were removed, not only reducing the extent of the anomalies in the eastern part of the study area but also increasing the strength of the local anomalies in the western part of the study area, highlighting the weak anomalous information that is hidden in the low-background area in the western part of the study area. At the same time, the known deposits (points) are located near the high-value areas of the anomalies. Combined with the geological conditions of the study area (Figure 2), the background map of the decomposed PC2 principal component (Figure 8d) shows that the background area is controlled by fractures and intrusive rocks. The high-background anomaly is located above the Late Carboniferous granitic mylonite; the decomposed anomaly (Figure 8c) is closely related to the rocks, and the distribution of the decomposed PC2 principal component anomaly is somewhat similar to that of the PC1 principal component anomaly in the middle- and high-anomaly areas; both are distributed near known Au deposits (points) and have some spatial correlation. This further illustrates the anomalous mineralization

information reflected in the PC1 and PC2 principal components, which are dominated by Au mineralization. Regarding the plotted lnS-lnA($\geq$S), the slope of the fitted curve reflects the different self-similarity characteristics. At the same time, the energy spectrum distribution in the study area is linear, which reflects the fact that the anomalies distributed in the study area are self-similar in the frequency domain and belong to the same fractal distribution, with a high probability of being the products of the same process, further demonstrating that the anomalies and background fields extracted by the S–A method are more consistent with actual geological conditions and can more effectively indicate the location of occult deposits.

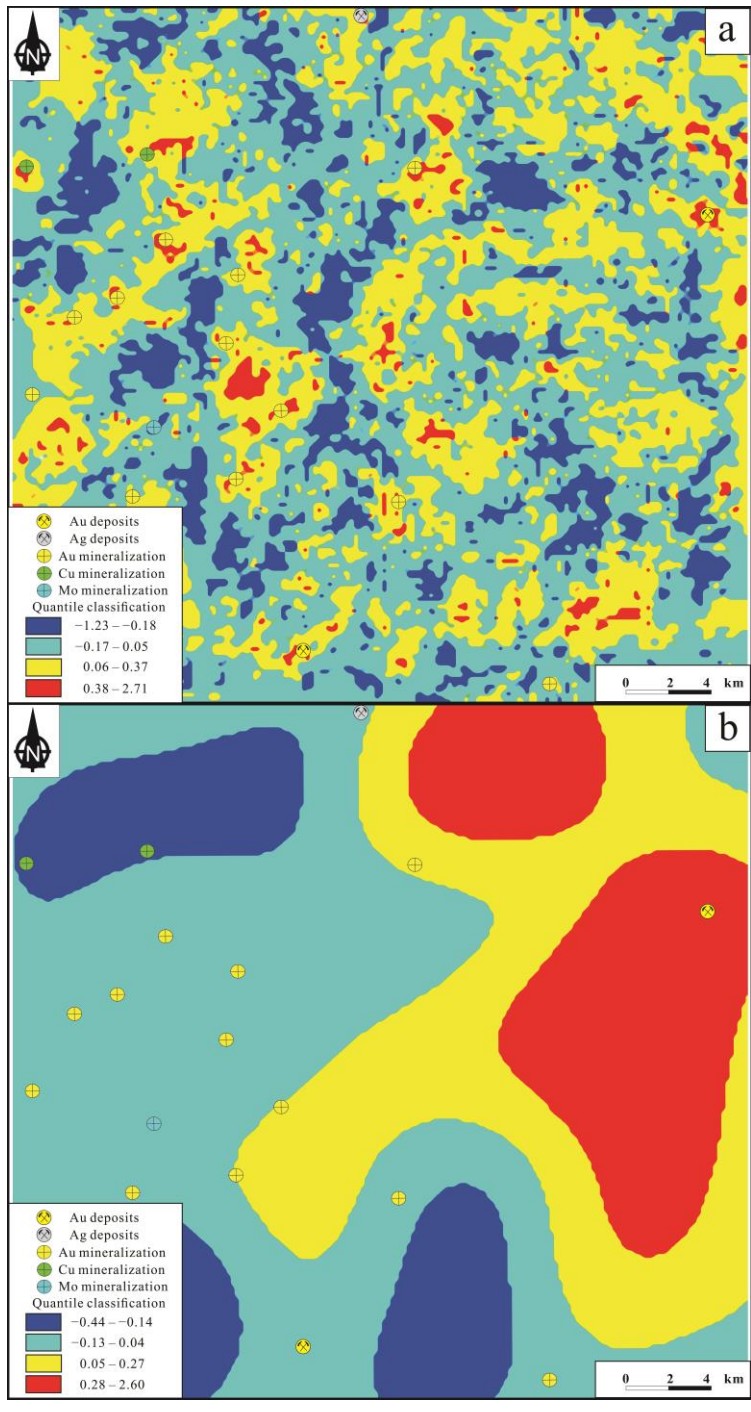

**Figure 8.** *Cont.*

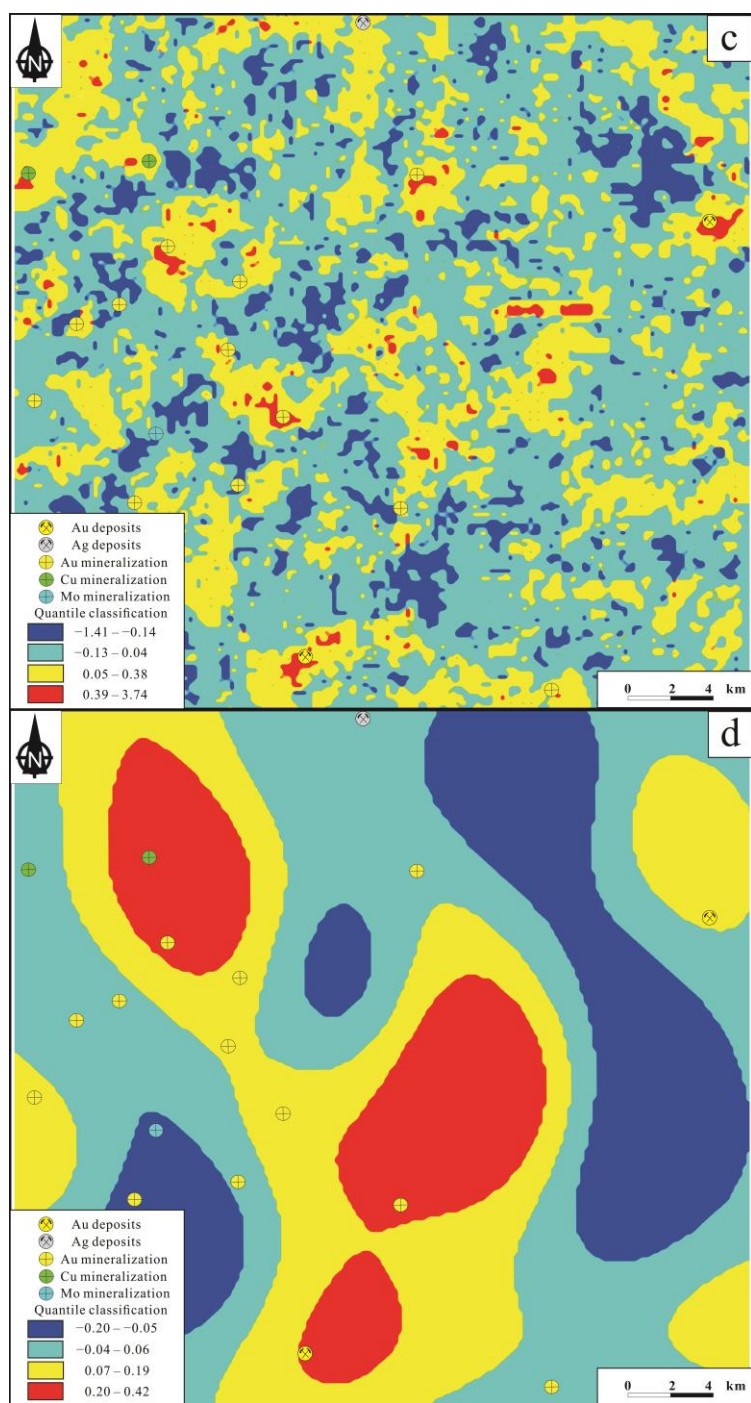

**Figure 8.** The decomposed maps for the anomaly map (**a**) and background component (**b**) of the PC1 component. The decomposed maps for the anomaly map (**c**) and background component (**d**) of the PC2 component.

## 5. Conclusions

1.  Geochemical data are typical compositional data with a closure effect. Before the data can be statistically analysed, an ILR-transformed of the data is required. This method can effectively eliminate closure effects in geochemical data while revealing the true spatial distribution pattern of elements.
2.  The PC1 and PC2 principal components associated with mineralization were obtained by robust principal component analysis of the ILR-transformed data from the study

area. The PC1 and PC2 principal components reflect a combination of elements associated with Au mineralization.

3. The S–A method takes into account the spatial geometry and frequency distribution of geochemical patterns. It provides an effective means for characterizing geochemical anomaly fields and decomposing diverse geochemical fields.

4. The S–A method was used to decompose the composite anomalies of the PC1 and PC2 principal component combinations in the study area, and the decomposed anomalies and background information were in good agreement with the known Au deposits (points). At the same time, a number of geochemical anomalies with prospecting potential were obtained in their periphery, which provided a theoretical basis and exploration focus for the next instance of ore prospecting and exploration in the study area.

**Author Contributions:** Conceptualization, K.Q. and Z.Z.; methodology, K.Q. and Z.Z.; software, K.Q. and Y.L.; validation, Z.Z., K.Q. and J.C.; formal analysis, K.Q. and Y.L.; investigation, C.L.; resources, Z.Z. and C.L.; data curation, K.Q.; writing—original draft preparation, K.Q.; writing—review and editing, K.Q. and Z.Z.; visualization, K.Q., Y.L. and J.C.; supervision, K.Q. and Z.Z.; project administration, Z.Z.; funding acquisition, Z.Z. All authors have read and agreed to the published version of the manuscript.

**Funding:** This work was supported by the Project of the Natural Science Foundation of Liaoning Province (2020-BS-258), and the Scientific research fund project of the educational department of Liaoning Provincial (LJ2020JCL010). The project was supported by the discipline innovation team of Liaoning Technical University (LNTU20TD-14) and the Key Research and Development Project of Heilongjiang Province (GA21A204).

**Acknowledgments:** Thanks to the chief editor and reviewers for their review and constructive comments, which have played a great role in the improvement of this paper.

**Conflicts of Interest:** The authors declare no conflict of interest.

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
