# Peer review of "Geochemical Data Mining by Integrated Multivariate Component Data Analysis: The Heilongjiang Duobaoshan Area (China) Case Study"

_minerals, doi:10.3390/min12081035_

Round 1
Reviewer 1 Report
Please find my review of the paper entitled "Geochemical data mining by integrated multivariate component data analysis. The Heilongjiang Duobaoshan area (China) case study" by Zhao et al.
The manuscript reports a new and exhaustive geochemical study of soil samples from Heilongjiang Duobaoshan area (China). The further data processing by multivariate component data analysis method is also thoroughly described.
However, there is a lack of the geochemical modeling and the interpretation of the results. The conclusions are not informative about the features or generation processes of the area. I'm not sure what is the purpose or this research.
The figures are not intuitive and are difficult to understand.
It is needed a refocusing of the “Results and Discussion” and “Conclusions” sections.
Please, see below the minor comments or suggestions keyed to line numbers:
Line 159. What is the B layer?
Line 161. Sub-clay, et al., …. I don’t understand
Line 183. Please define the concept of closure effect.
Line 211-213. I don't understand what means this sentence. Please rewrite.
Line 236. This is a data processing but... Where are presented the data?
Line 291. Au is a favorable factor for mineralization… What does this means?
Line 308. The image has very low resolution.
Line 343. Basalt not basal.
Best regards,

Reviewer 2 Report
The manuscript presents and discuss a novel technique of analysis and interpretation of geochemical data able to extract more information from a given set of data.
I have found this approach of interest, which worth publishing.
Some nonessential remarks can b found in the attached annotated file.
Beside these, I would ask the manuscript to be thoroughly revised by a native speaking English. Some phrases are really to long to be understood at the first reading

Reviewer 3 Report
Dear authors, The article titled “Geochemical data mining by integrated multivariate component data analysis. The Heilongjiang Duobaoshan area (China) case study ” written by Hai et al. is a short, precise, and reasonably well documented. The authors undertake the research on combined methods for geochemical exploration to obtain a true spatial pattern distribution of geochemical elements in a geological complexity of the Heilongjiang Duobaoshan area. The authors have drawn reasonable interpretations and conclusions. However, the following changes need to be incorporated into the manuscript to bring it to publication in the MINERALS journal. For sample collection of soil should be mention if it is a compositional sample and if the total weight of 150 g (taking into account the grain size of soil) is a representative quantity. To each analysis method, a short description of sample preparation is needed (or a direct reference if it is presented elsewhere). Also, detailed about the certified reference materials are missing. Table 1 present the detection limit in ppm, please change these units because are ambiguous. Figure 9 present c, d not a, b respectively. Good luck !
Round 2
Reviewer 1 Report
The manuscript has been modified according to the suggestions. Some additional comments:
Line 305: Au is a favourable factor... Please rewrite this sentence. Something like... Au is the main-ore forming metal...
"Conclusions" section can be improved with a more intuitive and concrete writing.
